# Organomorphic Carbon Preform Formation Mechanism

**Evgeny Bogachev** 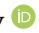

JSC Kompozit, 141070 Korolev, Russia; eug-bogatchev@mail.ru; Tel.: +7-495-513-2306

**Abstract:** Looking for ways to increase the structural uniformity of ceramic matrix composites (CMC) resulted in the development of organomorphic composites (C/C, C/SiC, SiC/SiC) where the filament diameter is comparable to the space between the filaments. The structural uniformity of the aforesaid CMCs is determined by their reinforcing preform; however, the mechanism of formation of this structure from polymer fibers remains unclear. This paper discusses an investigation of pressed specimens of the OKSIPAN® nonwoven fabric based on Pyron® polyacrylonitrile (PAN) fibers that were underoxidized as was determined using the electron paramagnetic resonance and microtomography methods. Using electron scanning microscopy, thermomechanical analysis and X-ray tomography, cementation of the preform due to the release and condensation of readily-polymerizing resin-like substances on the fiber surface after pressing at 180 °C was shown to be mainly responsible for retaining the mutual positions occupied by the fibers during pressing. The carbonized residue of the resin-like substances binds the fibers after pyrolysis. The other reason for organomorphic carbon preform consolidation is autohesive interaction of insufficiently cross-linked cores of the PAN fibers, since their thermal oxidation during pyrolysis at up to 1000 °C is hindered by the relatively high density of the compressed polymer preforms. The combination of pressing, thermal stabilization and pyrolysis results in the formation of the organomorphic carbon preform that features a relative density of at least 0.3 and a collection of pores, their normalized diameter ranging between 4 and 40 μm.

**Keywords:** polyacrylonitrile (PAN); fiber; pressing; pyrolysis; organomorphic carbon preform (OCP)

## 1. Introduction

The progress in ceramic matrix composite (CMC) production technology inspired the developers to compare CMCs with metals and traditional ceramics. Although showing significant advantages over these materials when it comes to the operation temperature (as compared to metals) and crack resistance (as compared to ceramics), CMCs are significantly inferior to them in their structural uniformity. Simply said, neither foil nor thin glass can be made of CMC, since the size of the CMC structural inhomogeneities turns out to be comparable to their thickness. Of course, when it comes to CMC, it is basically impossible to speak about a very thin part like a foil, since the diameter of the smallest CMC structural element—the reinforcing filament—ranges between 7 and 15 μm. However, the fabrication of CMC where the fibers are almost equispaced at 10 to 30 μm would make it possible to produce articles with a minimum thickness of 150 to 200 μm and uniform distribution of their filaments and matrices. Such an opportunity could expand significantly the area of CMC application including various graphite application scenarios, conventional ceramics and, of course, metals.

It seems clear that the size of the CMC minimum structural unit is determined by the preform. Almost all CMC preforms are machine-fabricated, and the complex thread composed of at least 1000 filaments, given the process specificity, determines the non-uniformity of the pore gaps. While in a complex thread the pore size between the filaments is of order of tenths of a micron, the distance between the filaments can reach hundreds of microns. For the Naxeco-type needle-punched preforms [1], the problem of achieving the required porous structure uniformity has not actually been solved as well.

The drive to achieve the minimum pore size, i.e., close to the filament diameter, throughout the entire preform volume, arose from recent studies focused on the creation of so-called organomorphic CMCs [2–4]. The initial material for them is neither carbon nor ceramic fibers, but their precursors, namely, organic polymer fibers. The main method for controlling the distance between the fibers involves the pressing of a precursor fiber package, followed by pyrolysis in a pre-stressed state. Specifically, the stable distance between the carbon fibers within organomorphic carbon preform (OCP)—about 10–40 μm –makes it possible to fabricate thin (0.3–0.4 mm) radio-frequency ion thruster electrodes (instead of molybdenum ones) from a carbon–carbon composite showing a high degree of surface purity [5], as well as to replace the graphite chase of the hot pressing mold with an OCP-based carbon–carbon one [6].

This paper addresses the mechanism of OCP formation from PAN fibers. The peculiarities of PAN processing to obtain a complex carbon filament have been thoroughly studied [7–9]. For polymer structure transformation to produce condensed pyrolysis products, the PAN fibers are subject to thermal stabilization by cyclization of the macromolecules that involves heating up to 290–350 °C in air, whereupon they are carbonized at a temperature of 1000–1200 °C. With the well-proven multifilament complex PAN filament pyrolysis processes, it is possible to obtain extremely strong (with the tensile strength of up to 5 GPa or more) carbon fibers.

The PAN pyrolysis involves the release of gaseous products (hydrogen cyanide, ammonia, water vapor, methane and carbon oxides), as well as noticeable amounts (more than 7% by weight) of resin-like substances, among which, according to the classical research results [10], are acrylonitrile, vinyl acetonitrile, pyrrol, acetonitrile, butylonitrile and propylonitrile. From practical experience it is known that these readily polymerizing compounds, when released from continuous PAN fibers at the pyrolysis stage, settle on the process equipment exhaust pipe surface as an undesirable sediment. It is clear that, given the enclosed space conditions experienced by the pressed preforms made of PAN fibers, one should expect them to have a noticeable effect on the state of the OCP being formed.

## 2. Materials and Methods

As the initial material for OCP preform formation, non-woven fabric OKSIPAN® (TR 8397-002-45680943-2010) from Niagara LLC (Shchelkovo, Moscow region, Russia) featuring a surface density of 230 g/m² was used. The web consisted of medium-twist staple fibers (cut into 50 to 60 mm segments) of PAN oxidized Pyron® (Zoltek Corporation production, Nyergesujfalu, Hungary), their linear density being equal to 2.2 dtex.

The studies were performed for three types of polymer preforms:

- Monolayer OKSIPAN® (M). For the experiments, the OKSIPAN® monolayer was pre-treated with thin (150 μm) water jets using the Spunlace technology [11] (MS specimen).
- Multilayer preform consisting of 60 MS monolayers (2D-MS specimen).
- Multilayer preform consisting of 60 M monolayers stitched by transverse needle punching (2D-M specimen), with a density of 0.15–0.2 g/cm³. Since in this spunlaced web, the threads are entangled with each other and it is impossible to pull them out during needle punching for stitching the preform, for this type of preform, the web specimens were not treated with water jets.

All polymer preforms 200 × 200 mm in size were compacted by applying force across the layers using a 1000 kN hydraulic press at a temperature of 150 °C followed by cooling together with the press. Immediately after pressing, the specimens were subject to long-term annealing (30–40 h in total), while in a pre-stressed state, including thermal stabilization (at up to 290 °C in air) and carbonization in a resistance furnace (at up to 1000 °C in a non-oxidizing environment) [12,13]. The temperature in the furnace was measured using chromel/alumel thermocouples.

The oxidation degree (concentration of paramagnetic centers) of the initial PAN fibers was analyzed by the electron paramagnetic resonance (EPR) method using the original test equipment of NIC Uglekhimvolokno (Mytishchi, Moscow region, Russia). A quartz

vessel, 3 mm in diameter, filled with the sample fiber up to a height of about 1 mm was placed in the resonance chamber of the instrument. When the resonance conditions were reached, the EPR spectra were recorded, by which the concentration of the paramagnetic centers was determined to estimate the PAN fiber oxidation degree. The initial PAN fibers were also analyzed with the Skyscan-2011 nanotomograph using the following scanning parameters: U = 50 kV, I = 200 μA, rotation angle—0.3 °, averaging over 5 frames, spatial resolution—0.3 μm/pixel.

The pore size distribution in the initial web as well as in the pyrolyzed specimens was determined by the standard contact porometry method [14] using as a standard contact porometer a Porotech 3.2 (Newmarket, Ontario, Canada). With this method it is possible to study the specimen porosity over a wide (1 to $10^5$ nm) pore size range, while not destroying the material itself. Specimens 20 mm in diameter, 5 mm thick, cut out of carbonized preforms were examined. The porosity assessment was made on the assumption that all the pores are cylindrical.

In case of steady-state capillary equilibrium, the following is true:

$$P \cdot S = \Pi \cdot \sigma \cdot \cos \theta \tag{1}$$

where *P*—external pressure, *S*—pore cross section, Π—pore perimeter, $\sigma$—liquid surface tension, $\theta$—wetting angle.

For the cylindrical pore model, the parameter characterizing the pore size is the characteristic radius:

$$r = 2\,\sigma \cdot \cos\,\theta/P. \tag{2}$$

The PAN fiber behavior under load was analyzed by the thermomechanical analysis (TMA) method using a TMA Q400 EM (New Castle, DE, USA). The specimen, in the form of a MS monolayer package 7 mm in diameter and 3.4 mm thick, was placed in a special container, 7 mm in diameter and 11 mm high, with a wall thickness of 1.5 mm, made of high-density carbon–carbon composite. The package was then pressed with a quartz indenter (Figure 1) and heated up to 350 °C in air at a rate of 1 deg/min applying a static load of 0.588 N and a dynamic load of ±0.294 N, while recording the time-dependent changes in the specimen's elasticity modulus and thickness.

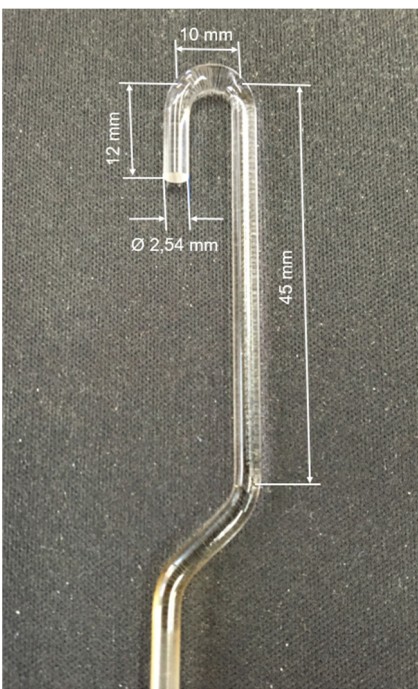

**Figure 1.** Indenter EXPANSION (general view).

After TMA, the package structure was examined using the X-ray tomography system XT H 320 LC from METRIS (Tring, Great Britain). The OKSIPAN® as well as 2D-MS sample were analyzed by IR spectroscopy using the Fourier spectrometer Jasco FT/IR-4600 (Tokio, Japan).

To study the structural changes in the material, the scanning electron microscopes JSM-7001F and JSM-6610 LV from JEOL equipped with an AdvancedAZtec energy dispersion analyzer were used. For the analysis, specimens about 20 × 10 mm in size were cut out of the preforms using a sharp knife.

To estimate the strength of the carbon filament in the organomorphic carbon preform, a unidirectional frame made of Pyron® PAN fibers 60 × 15 × 8 mm in size was subject to pyrolysis under a load of 0.1 MPa using a special graphite mold. After carbonization, carbon filaments about 10 μm in diameter, 40 mm long, were pulled out of the resulting OCP, and their ends were then pasted on to appropriate cardboard frames, whereupon they were subject to tensile tests with a load rate of 2 mm/min using a Zwick/Roell 5 kN ProLine test machine.

## 3. Results

### 3.1. Initial PAN Fiber Properties

Analysis of the paramagnetic center (PMC) concentration using the EPR method showed the initial PAN fiber to have a significantly lower degree of oxidation as compared to the fully oxidized fiber used in the production of continuous carbon fiber (Table 1).

**Table 1.** Comparison between the properties of the initial PAN fiber and the standard oxidized PAN tow used in continuous carbon fiber production.

| Item No. | Material | Hydrostatic Density, g/cm$^3$ | Paramagnetic Center Concentration [R]·10$^{17}$, Spin/g |
|:---:|:---:|:---:|:---:|
| 1 | Initial staple PAN fiber Pyron® | 1.37 | 4.7 |
| 2 | Oxidized PAN tow used in continuous fiber production | 1.39 | 23.0 |

These data correlate with the results obtained for the initial PAN fibers using a microtomograph, which revealed the presence of a denser near-surface region with a wall thickness of 1.5–2 μm (Figure 2).

This seems to be caused by the greater fiber oxidation degree of the fibers at the surface, which proceeds gradually from the outside to the inside. This phenomenon is common and can often be observed during oxidative thermal stabilization of PAN fibers [15,16].

Thus the results obtained using microtomography indicate different thermal stabilization degrees of the initial PAN fibers along the cross section. Absence of a thermally stabilized core determines the fiber behavior discussed below, as was observed during OCP formation.

### 3.2. Analysis of the PAN-Based Organomorphic Preform (OP) Properties during and after Pressing

3.2.1. Analysis of the Deformation Characteristics

Figure 3 shows the thermomechanical curves obtained for the PAN fibers in the form of the MS layer package (2D-MS).

The above dependences reveal a large deformation potential of the PAN monolayer package that begins to condense, reducing its thickness, from the very beginning of the heating process. At this stage of deformation, the volume of the interlayer gaps is reduced as the fibers come close to each other. After cooling, the layer package bears the marks of pronounced irreversible deformation at the point of indenter impact (Figure 4).

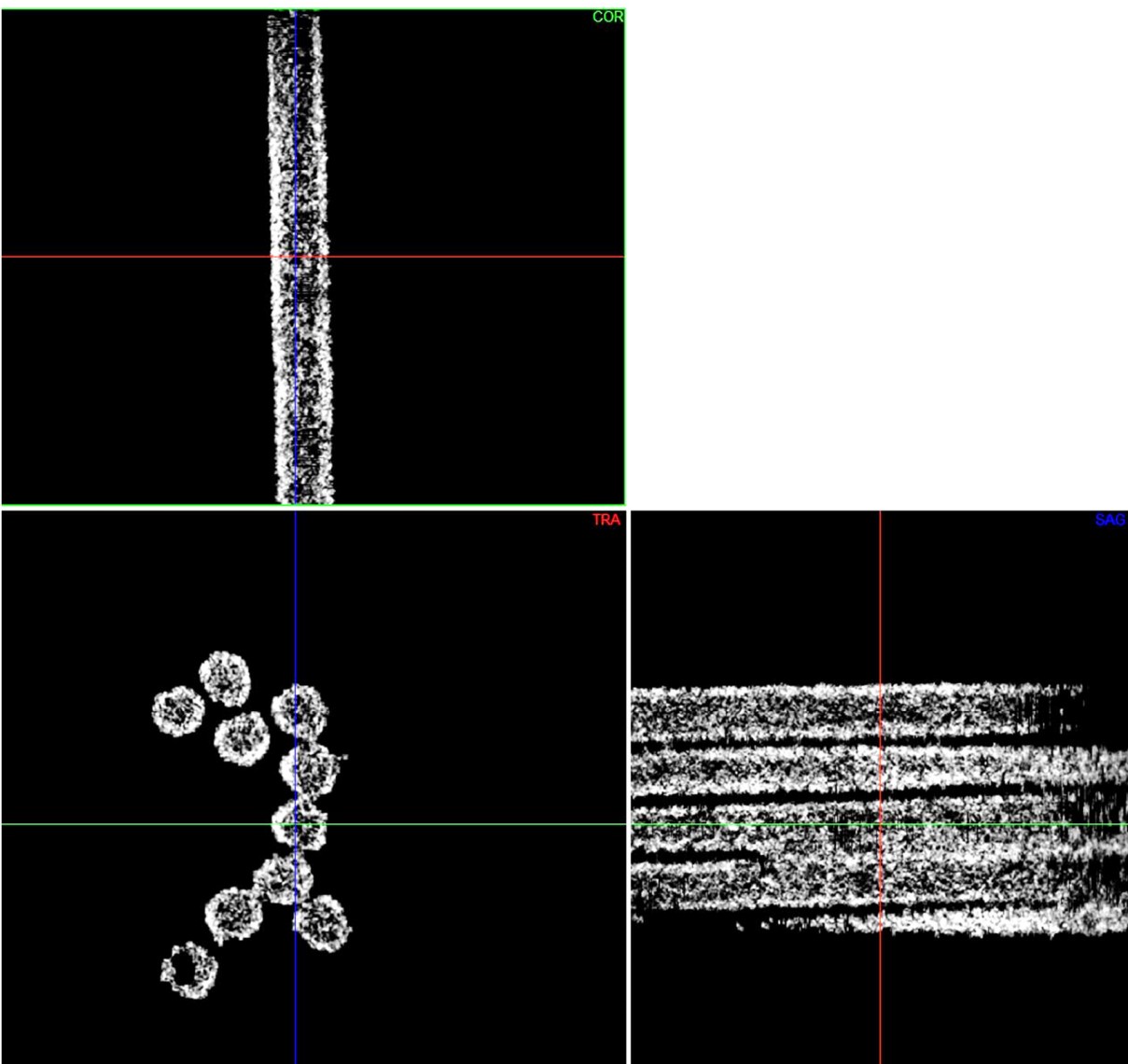

**Figure 2.** Tomographic images of the initial PAN fiber.

At the point of deformation, the package thickness is equal to 2.8 mm, with its initial thickness being equal to 3.5 mm, which is not incommensurate with the specimen thickness decrease under the indenter (see Figure 3).

Analysis of the 2D-MS specimen composition after TMA showed that the chemical composition did not change, but the intensity of wave signals was decreased (Figure 5). This testifies to oxidation or internal cross-linking of the 2D-MS sample.

This is also evidenced by a sharp increase (from 6–7 MPa to 27–28 MPa) in the elasticity modulus of the 2D-MS sample starting from a temperature of 250-270°C (see Figure 3), and in this case, the sample curing can be caused not only by the structural rearrangement in the PAN fibers. The presence of the resin-like substances within the pressed specimen, their boiling point ranging between 80 °C and 240 °C, results in a kind of gluing of the fibers after cooling. These readily polymerizing substances [10] can also contribute to preform cementation, when heated to 350 °C in air.

### 3.2.2. Analysis of the OP and OCP Microstructure

Analysis of the PAN fiber pressed preform microstructure showed the resin-like substances start being released from PAN fibers as early as at the pressing stage (Figure 6).

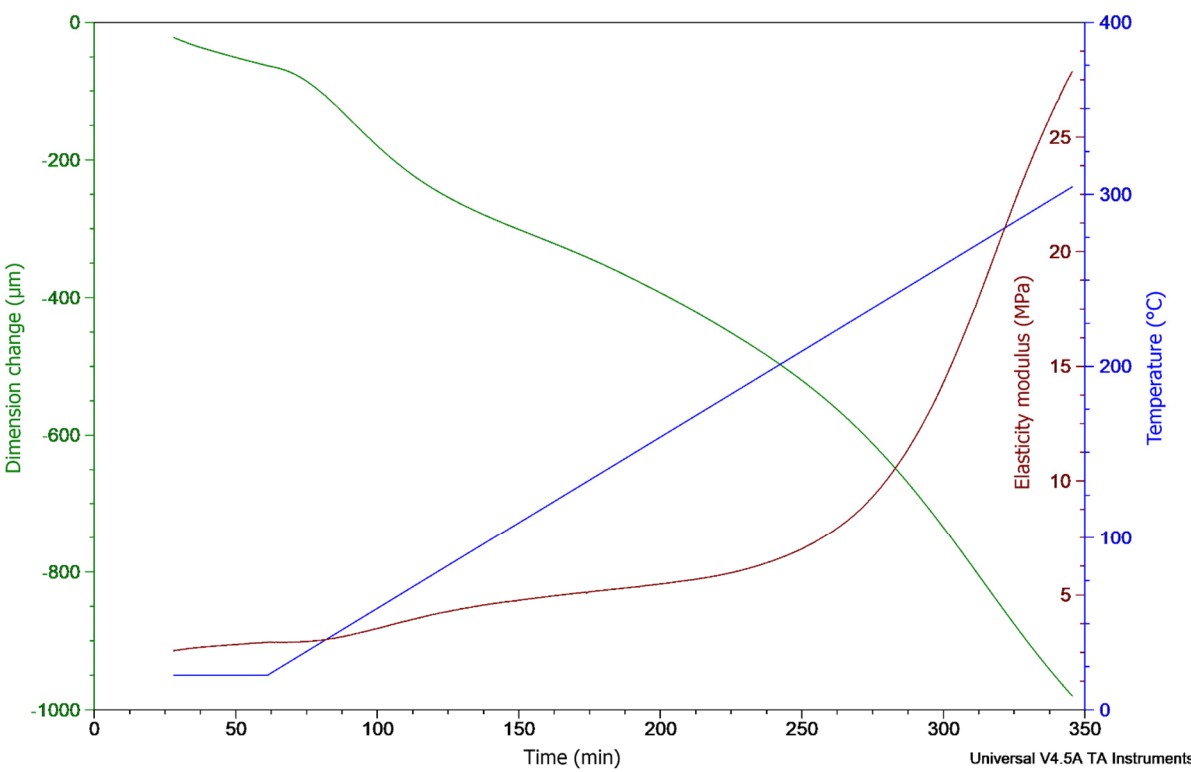

**Figure 3.** Temperature dependence of the elasticity modulus and dimensional change obtained for the 2D-MS specimen.

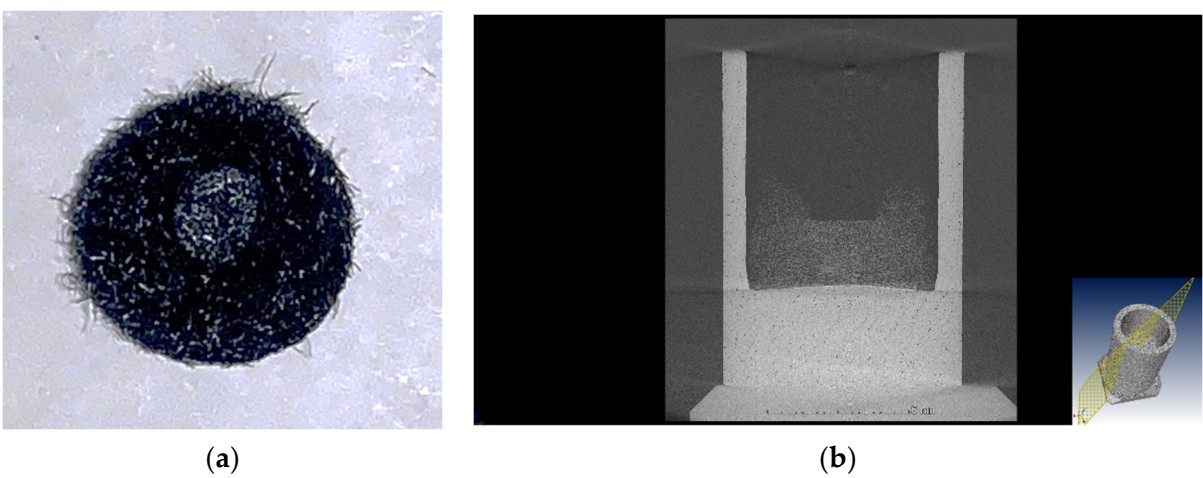

(**a**)            (**b**)

**Figure 4.** 2D-MS specimen after TMA up to 350 °C: top view (**a**) and tomographic image (**b**).

It can be seen that after thermal stabilization in air, the fibers are cleaned, and there are no more signs of resin-like substances (Figure 7).

However, during the experiments, after annealing in air at 290 °C, the pressed preform thickness was found not to change with time. This can also be explained by the fact that almost all substances released during pressing and then during heating tend to polymerize [10]. During annealing, the liquid and gaseous resin-like substances travel intensively within the preform while polymerizing. The thin film of cured polymers on the fiber surface, especially at their contact points, is able to bind the fibers in the preform. In this case, the preform thickness achieved under pressing is frozen due to the carbonized product of the resin-like substances (Figure 8).

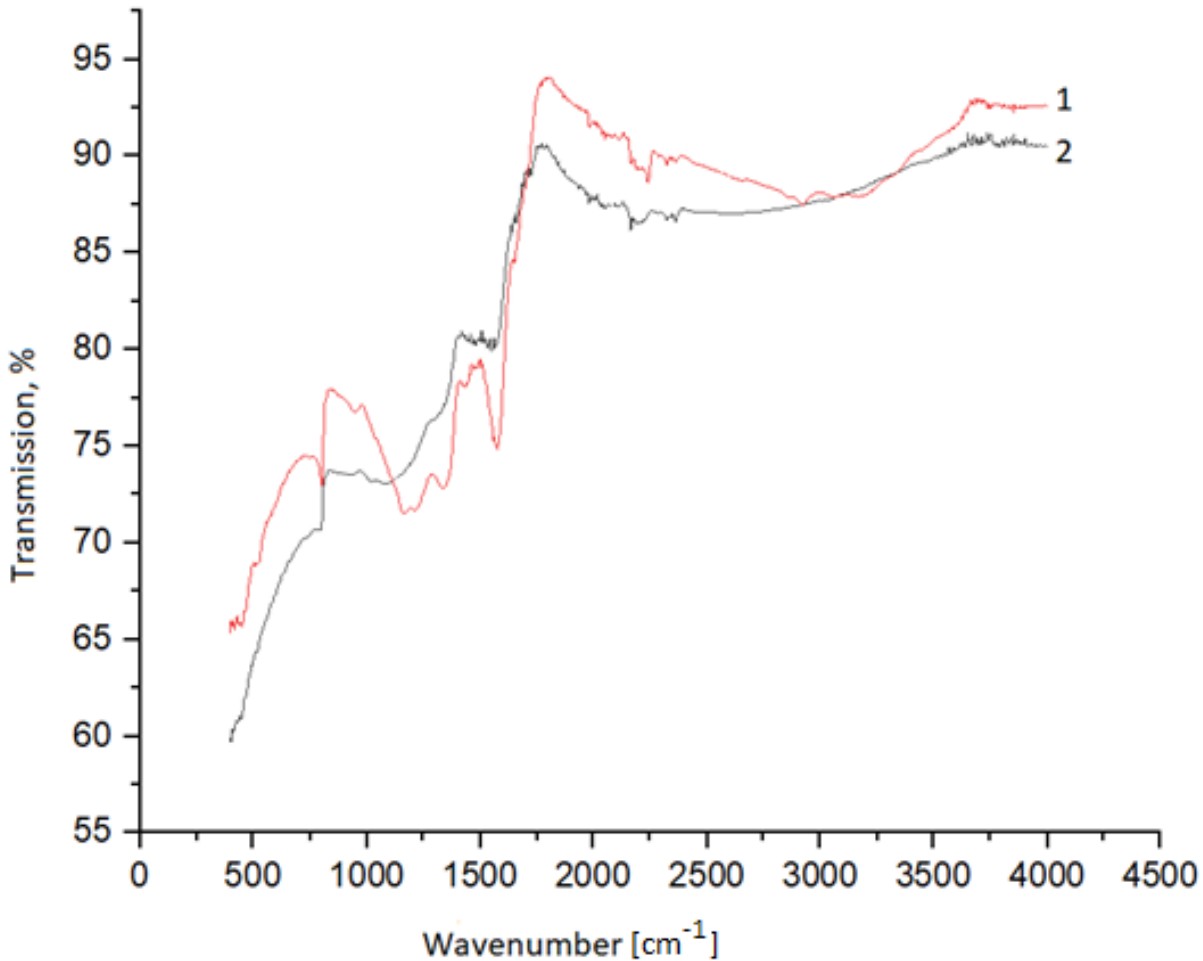

**Figure 5.** Results obtained for the OXIPAN® (1) and 2D-MS specimen after TMA (2) using IR-spectroscopy.

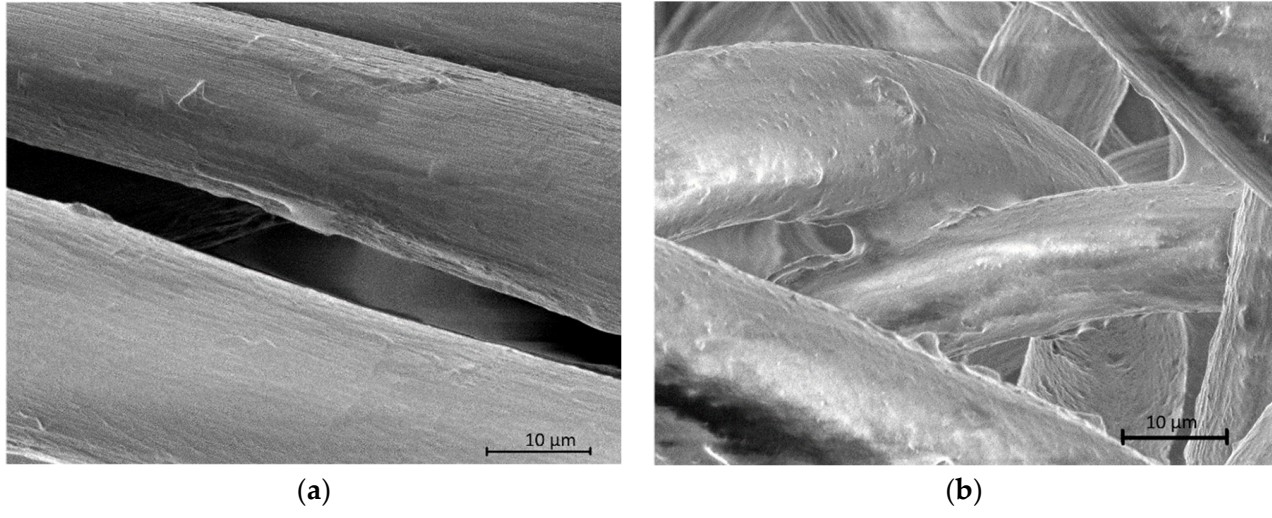

(**a**) (**b**)

**Figure 6.** PAN fiber microstructure (**a**) before and (**b**) after pressing of 2D-M specimen.

The PAN fiber package compaction by pressing that governs the phenomena occurring within the preform manifests itself in the form of the plastic deformation of the polymer fibers as well. This deformation is most clearly exemplified by the compressed carbonized

MS specimen (Figure 9) where, in addition to filament bending, numerous deformations of the fibers under pressing can be seen.

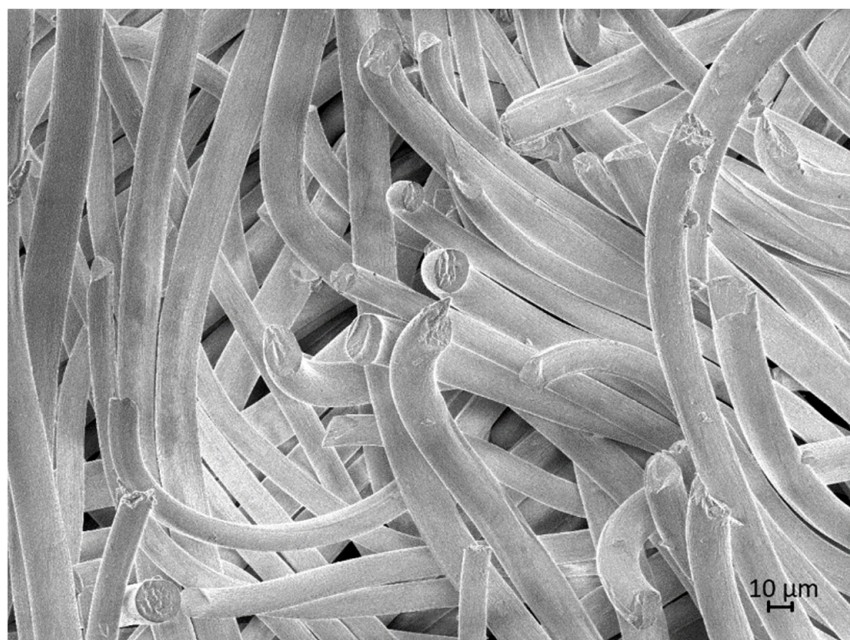

**Figure 7.** Pressed of 2D-M specimen after annealing in air at 290 °C (general view).

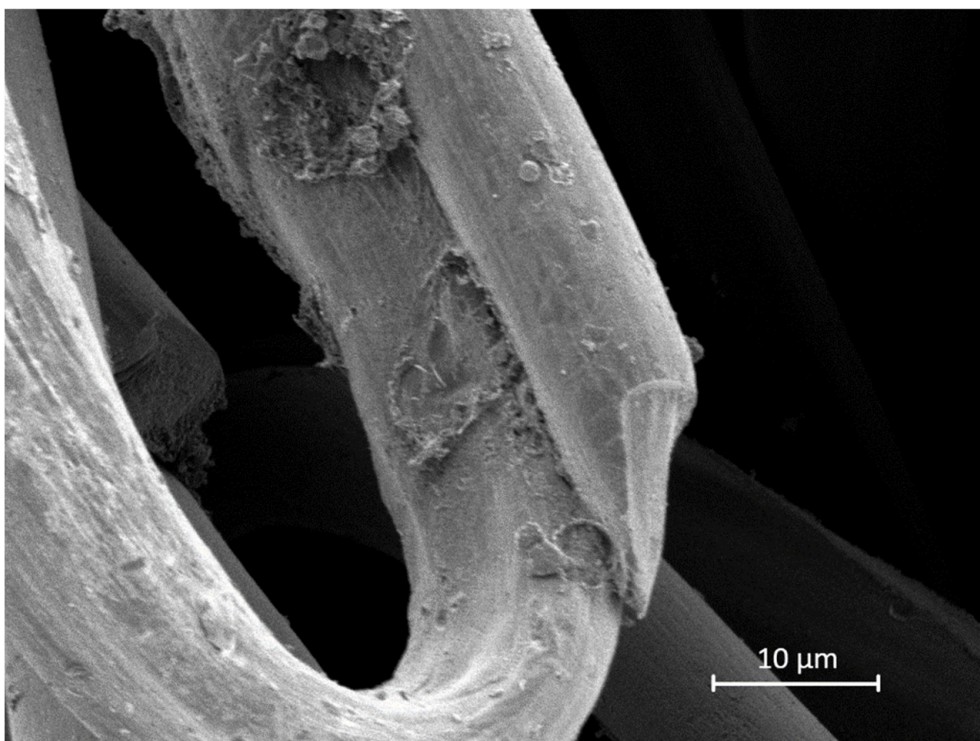

**Figure 8.** Deposit of the carbonized resin-like substances on the OCP surface.

Along with the obvious highly elastic compressive deformation, there are regions in the PAN fiber monolayer that attest to a viscous flow of the polymer during pyrolysis. Apparently, the polymer did not have enough time to stabilize just outflows, in the literal sense of the word, from the fiber core, thus binding the filaments together (Figure 10). As a rule, these regions are adjacent to the ends of the broken fibers.

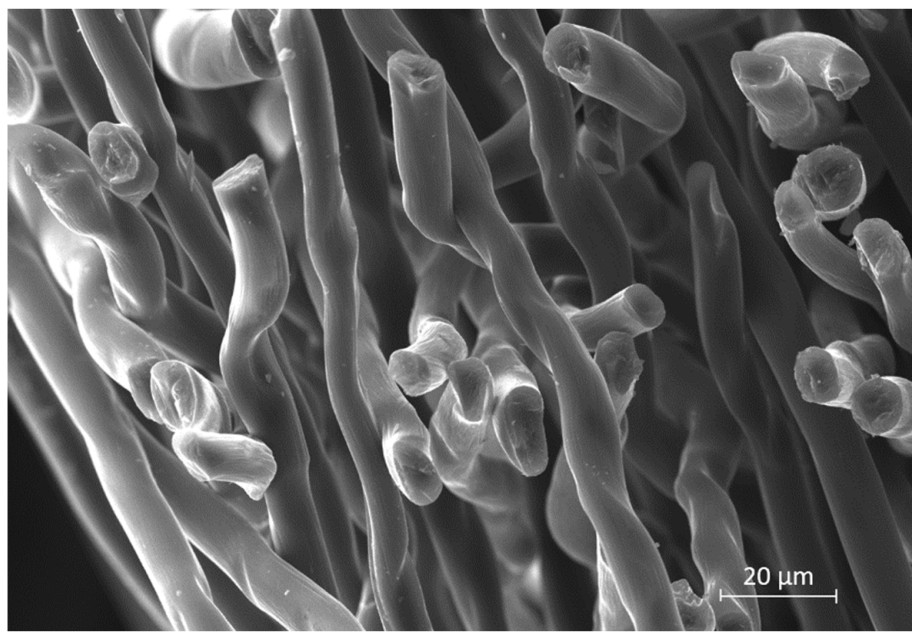

**Figure 9.** Microstructure of MS specimen that was subject to pressing, after carbonization.

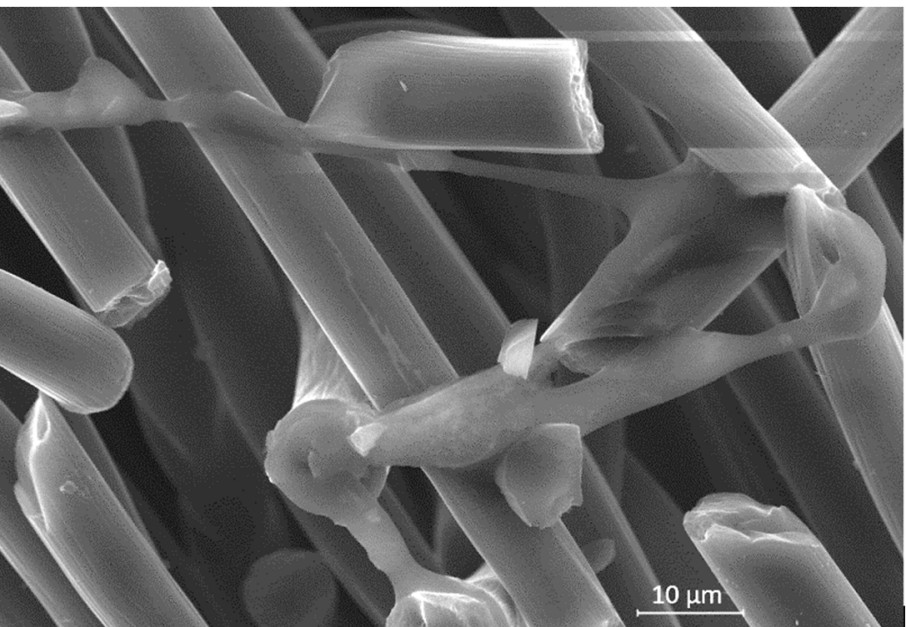

**Figure 10.** Viscous flow in MS specimen (example).

After pressing and pyrolysis, the 2D-MS specimen also shows signs of interaction; however, these signs can be observed more often and they are characterized by large-scale fiber splicing (Figure 11).

The microstructures shown in Figures 10 and 11 cannot be attributed to the influence of the resin-like substances. The observed bonding of the filaments is explained by the autohesive interaction of the macromolecules of the pressed PAN fibers under pre-stressed conditions and subsequent pyrolysis in this very state [2]. Perhaps, in this case, PAN, the melting point of which, as is known, is higher than the decomposition onset temperature, does have enough time to show thermoplastic properties. This is facilitated by diffusion hindering for the oxidative thermal stabilization of the fiber cores that often appear to be less dense after pyrolysis than at the near-surface layer (Figure 12).

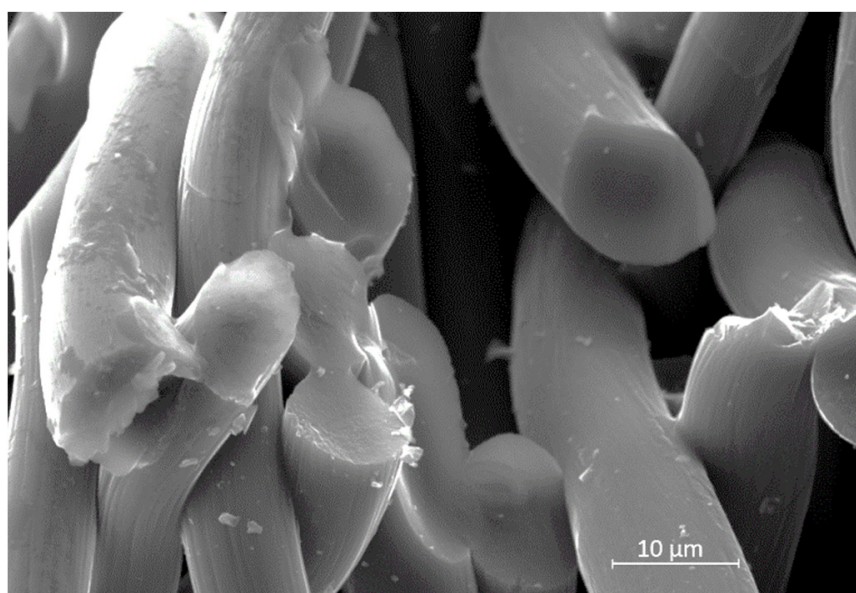

**Figure 11.** 2D-MS specimen microstructure after carbonization.

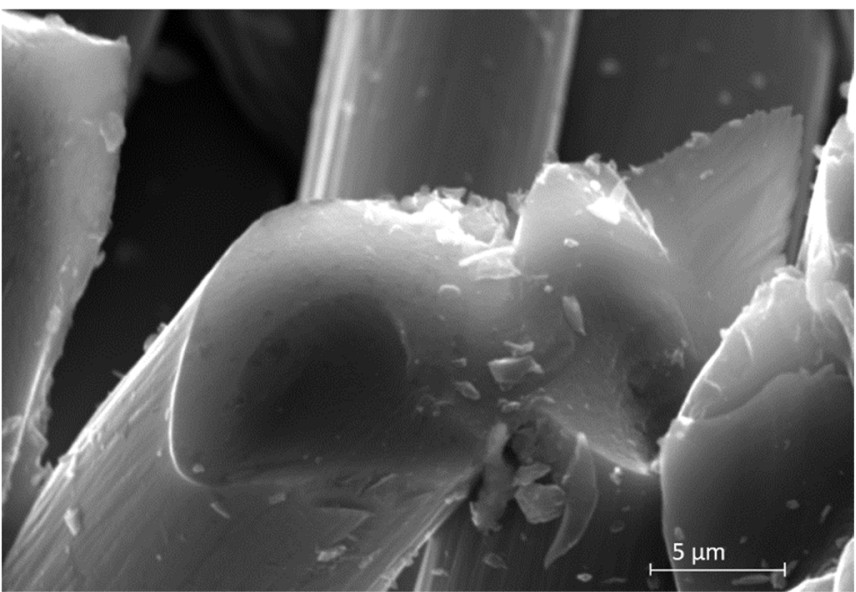

**Figure 12.** The areas with various densities within the OCP-fiber: at the fiber fracture surface, the less dense fiber core, about 6 μm in diameter of which is clearly visible.

Oxidation of porous materials depends on the conditions of oxygen delivery to the pores. In a massive specimen, hindered oxygen penetration to the fibers slows down the oxidation processes in the preform volume. When the pore sizes are small enough, while the porous body is sizable, oxidation can only take place in the near-surface region of the specimen. According to [17], there is a criterion for assessing the nature of porous body oxidation that appears as follows:

$$\omega = \frac{\rho_g \cdot d_m \cdot D_g \cdot c_g}{2K \cdot H},$$

(3)

where $\rho_g$—gas density; $d_m$—average pore size; $D_g$—oxidizing gas diffusion coefficient; $C_g$—gas concentration near the outer boundary of the porous material; $K$—oxidation rate constant; $H$—porous body thickness.

For the oxidation to proceed uniformly throughout the entire pore volume, ratio $\omega > 1$ must hold true; in this case, the pore surface oxidation rate is only determined by constant K. An increase in the thickness of the compressed polymer preform or a decrease in the average pore size reduces the likelihood of the effective air participation in the PAN fiber thermal stabilization process.

However, when examining the microstructure, it is relatively rare to observe autohesive fiber-splicing regions, so they are unlikely to be the main cause of the high strength of the pressed polymer fiber packages after carbonization.

### 3.2.3. OCP Properties

A distinctive feature of OCP is the high open porosity (no less than 60%) as well as the narrow range of pore sizes. Numerous studies made using the reference porometry method have shown the characteristic radius (in nanometers) of almost all voids in the preform to have a logarithm lg r value in the range of 3.5–4.5 (see, for example, [2]). Particularly, for the carbonized 2D-M preform (OCP Ipresskon® from JSC Kompozit, Korolev, Moscow region, Russia) (see Figure 13 for its general view and microstructure), the standart contact porometry showed a porosity of 63% with the main range (more than 90%) of the equivalent pore diameter of 4 μm to 40 μm (Figure 14).

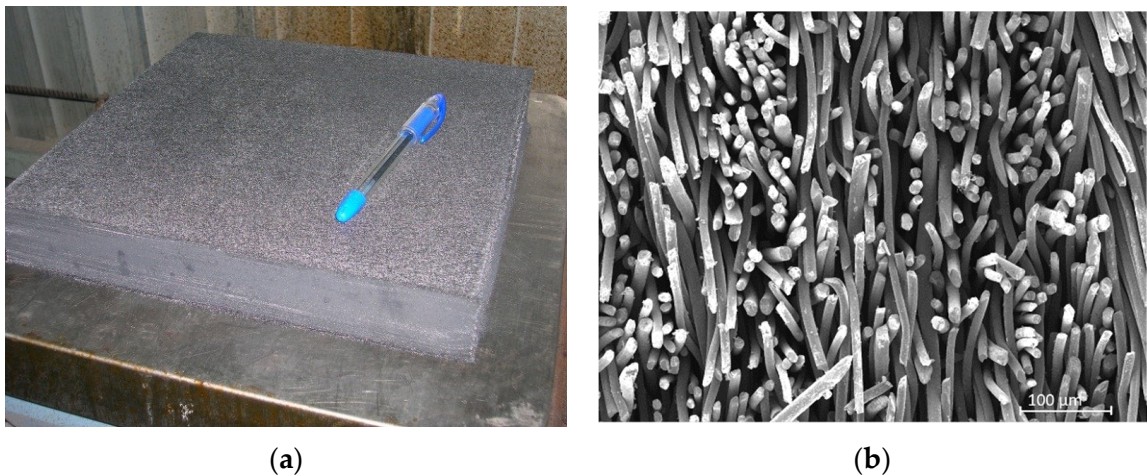

(**a**)            (**b**)

**Figure 13.** OCP Ipresskon®: general view (**a**) and microstructure (**b**).

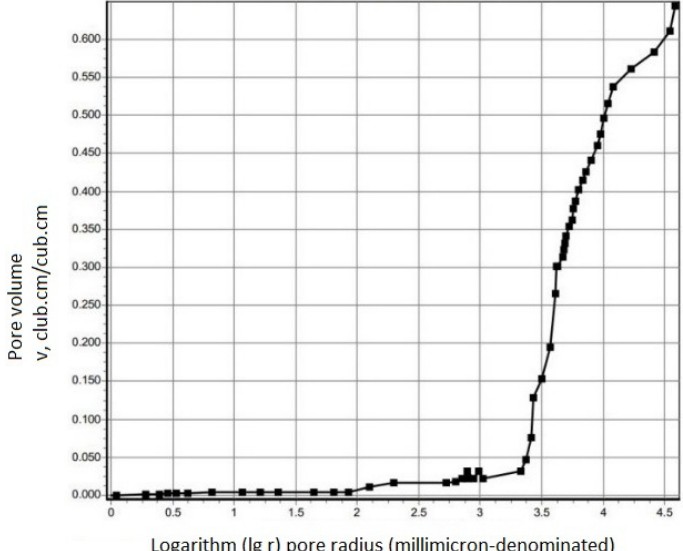

**Figure 14.** Integral pore distribution for OCP Ipresskon®.

Given the average carbon fiber density of 1.4 g/cm$^3$, the OCP density was found to be 0.58 g/cm$^3$, with the relative density being equal to 0.36. At this density, OCP represents quite a strong preform (see Figure 13a) featuring a uniform porous structure (see Figure 13b).

Certainly, the carbon density estimated based on the porometry analysis results is significantly less than that of the continuous carbon fibers, namely, at least 1.6–1.7 g/cm$^3$. However, according to (3), during thermal stabilization, it is more difficult for the air to penetrate to the inner layers of the polymer preform than to the near-surface ones. Air participation in the process of formation of the dense and strong PAN fiber structure is extremely important. Hence the density (and therefore the strength) of the filaments near the surface may differ from those within the OCM volume. Therefore, the fiber strength values must also be different, as evidenced by the data shown in Table 2.

**Table 2.** Mechanical properties of the OCP carbon fibers.

| Sample No. | Fiber Diameter, μm | Maximum Force $F_{max}$, N | Filament Tensile Strength $\sigma_f$, MPa | Maximum Deformation $\varepsilon$, % |
|:---:|:---:|:---:|:---:|:---:|
| 1 | 10.0 | 0.1559 | 1986 | 1.55 |
| 2 | 9.2 | 0.1294 | 1948 | 1.27 |
| 3 | 10.6 | 0.1233 | 1398 | 1.25 |
| 4 | 9.0 | 0.1262 | 1985 | 1.30 |
| 5 | 10.6 | 0.1269 | 1438 | 1.06 |

Unfortunately, it was not possible technically to separate the fibers lying closer to the surface from those lying in the depths.

Figure 15 shows the characteristic tensile diagrams obtained for the PAN fibers, which reveal a consistent tensile strength increase when proceeding from the polymer state to the carbon one.

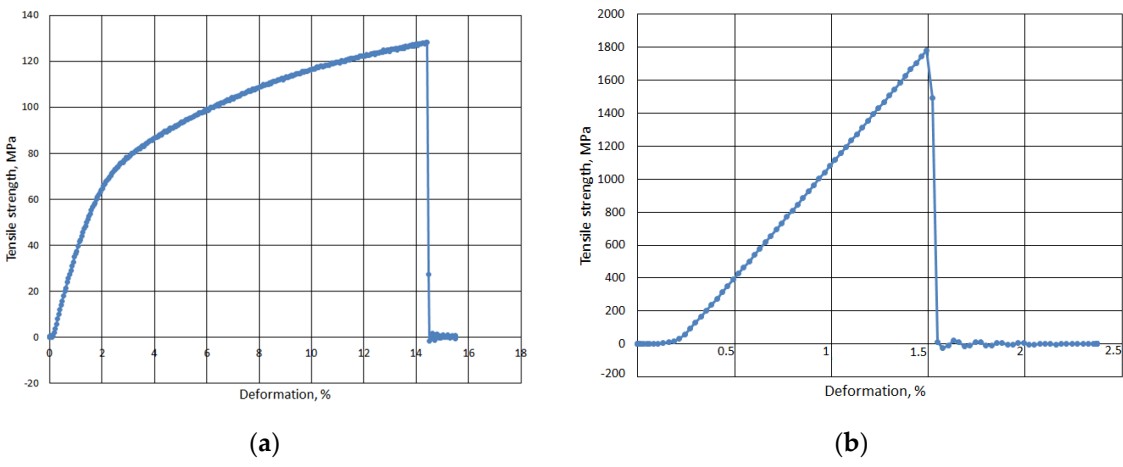

(a)  (b)

**Figure 15.** Characteristic tensile strength-strain diagrams obtained for the PAN fibers: (**a**) in their initial state, (**b**) after carbonization.

In Figure 12, the deformation starts from 0.25%, rather than from zero, due to a certain initial out-of-straightness of an individual filament when pasted onto the frame. Therefore, the force will remain zero until the filament straightens (when the holders of the test machine diverge).

The Table 2 data attest to a relatively low level and a significant variability of the OCP carbon fiber tensile strength values. However, the strength values obtained for the carbon–carbon composite [2,6] testify that the composite, when implemented based on OCP, can show quite high properties (tensile strength of up to 150 MPa).

The results obtained for 2D-M preform-based OCP reveal a direct dependence of the carbon yield of the preform fibers on the following:

- initial needle-punched plate density (0.15 g/cm$^3$, preferably 0.17–0.20 g/cm$^3$);
- the pressed preform thickness (plates 7 mm thick show a carbon yield of 64–65%, while those 12 mm thick—54–55%);
- thick non-pressed plates may provide a carbon yield lower than pressed but thinner ones.

## 4. Discussion

Being a key process when fabricating organomorphic carbon preforms, the pressing process transforms the conditions under which continuous carbon fibers have been produced worldwide for decades. The higher concentration of polymer fibers per unit volume promotes the influence of resin-like substance behavior during pyrolysis, as well as resulting in compressive deformations and giving rise to autohesion that is absent when using the conventional technology.

The monolayer organomorphic preform cementation mechanism is implemented taking advantage of the fiber flexibility: the thin water jets intertwist the fibers, thus making the monolayer much denser and stronger. When pressed, such a layer becomes slightly thinner just through reduction of the fiber thickness due to plastic deformation of their core. The role of the resin-like substances in the cementation of such a preform is zero; after carbonization, the MS specimen strip (as with the M specimen) has not lost its elasticity. Compression of the filaments is accompanied by stretching across the fibers, which breaks the side links of the macromolecules governed by the Van der Waals forces. They start losing the acquired preferential orientation of their macromolecules along the fibers, striving to line up perpendicular to the fiber axis [18]. It is probably for this reason that one cannot expect to see high mechanical properties of CMC obtained based on the MS monolayer organomorphic preform.

Quite different properties are demonstrated by the multilayer preforms, namely, 2D-M and 2D-MS. The phenomena occurring in them during pyrolysis can be summarized as follows. The MS strip retains its elasticity after pressing and carbonization. Although to a somewhat lesser extent, the two- and three-layer specimens, after pressing and carbonization, are able to bend. The four-layer specimen almost loses its elasticity (Figure 16).

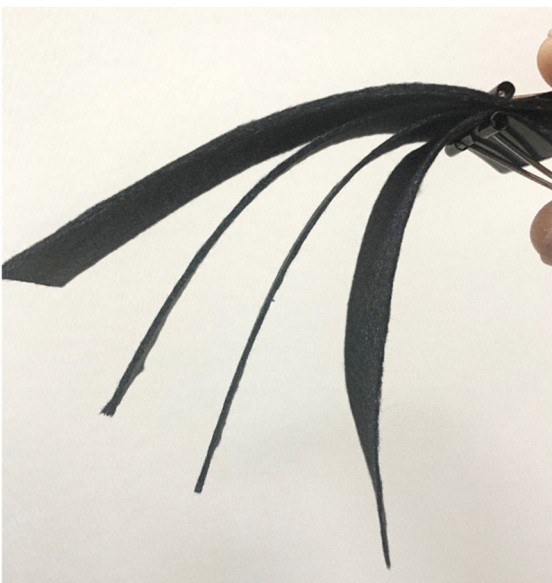

**Figure 16.** MS strips carbonized, when pre-stressed, with the number of layers increasing from one to four (from right to left).

Since the layers were not pre-bonded mechanically to each other (either by stitching or by needle punching), the multilayer specimen acquires its integrity through a kind of crosslinking of the layers, given the higher concentration of polymerizing volatiles and the interaction of the PAN fibers that have not been fully thermally stabilized. Given the hindered air penetration to the fibers due to the small pore size and the strong role the resin-like substances play in the preform volume, the lifetime of the undercured fibers in the preform turns out to be quite long. Due to the fact that the fibers in such a package are deprived of the ability to move freely, some regions of the package monolayers are subject to overpressure caused by the local non-uniformity of fiber distribution. Compaction of the package increases the number of contacts between the fibers [19]. The compressive stress is transmitted only through the contact areas. Therefore, the stress distribution non-uniformity seems inevitable, with the maximum stresses at the contact points and zero stresses between them. Where the fiber core is not sufficiently stabilized, such non-uniformity results in stress relaxation, including due to the formation of the autohesion contact. Therefore, in the stress-strain state under pressing, as well as given the hindered oxygen diffusion in the dense compressed preform, a viscous flow of the filament cores can be observed with the multilayer preform being bound into a single whole.

## 5. Conclusions

The OCP formation mechanism attests to the possibility of fabricating carbon fiber preforms showing controllable porosity that do not require any additional binders to fix the fiber positions occupied during pressing, including the minimum possible one. The pressing brings the filaments closer to each other, and this process is controlled by the pressing force. Therefore, it becomes obvious that it is possible to control the size of the voids between the fibers. Due to the release of resin-like substances, subsequent heating freezes the fibers in their occupied position. When close to each other, the fibers, due to the hindered diffusion (especially in a massive specimen), slow down the stages involved in the thermal stabilization process—namely, oxidation, cyclization and dehydration, and, given their close contact with each other, show autohesion interaction as was observed during microstructure analysis.

Two mechanisms of carbon fiber bonding in OCP, namely, the release of resin-forming substances from the PAN fibers and their autohesion interaction, contribute to the pressed polymer preform behavior as a cohesive whole during pyrolysis. This takes place despite the shrinkage of the preform (that may range between 8% and 17–20%, depending on the direction) when it passes to the carbonized state. Thus, the presence of a kind of "binder" prevents shear stresses during the preform transition from the polymer state to the carbon one.

To increase the carbon fiber strength in OCP, it is necessary to optimize the mass transfer conditions at the initial stage of pyrolysis, namely, during thermal oxidation.

**Funding:** This research received no external funding.

**Institutional Review Board Statement:** Not applicable.

**Informed Consent Statement:** Not applicable.

**Data Availability Statement:** Not applicable.

**Acknowledgments:** The author extends his great appreciation to his colleagues for their contribution to the research: the Director of NIC Uglekhimvolokno, M.B. Radishevsky and the Chief of Group JSC Kompozit, A.B. Elakov.

**Conflicts of Interest:** The author declare no conflict of interest.

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
