# Peer review of "Organomorphic Carbon Preform Formation Mechanism"

_jcs, doi:10.3390/jcs6020050_

Round 1

Reviewer 1 Report

Dear author,

the paper tells about micromechanical aspects which could well fit with the journal main themes.

I suggest the following actions for improving generally the manuscript quality:

  • the experimental activities shall be better presented (techniques, protocols, algorithms etc);
  • the figure 13 is not clear (what are the measure units? Are you confident that y-axis is strain?);
  • the stress relaxing mechanism should be much more detailed;
  • what is the effect on the shear stress of thermal load?
  • can you provide the material allowable in terms of MPascal?
  • the first person (we, I...) should be avoided.

Author Response

Dear author,

the paper tells about micromechanical aspects which could well fit with the journal main themes.

Dear Reviewer 1, thank you very much!

I suggest the following actions for improving generally the manuscript quality:

the experimental activities shall be better presented (techniques, protocols, algorithms etc);

Amendments have been introduced in Section 2.

the figure 13 is not clear (what are the measure units? Are you confident that y-axis is strain?);

Figure 13 (now Figure 15) has been updated.

the stress relaxing mechanism should be much more detailed;

what is the effect on the shear stress of thermal load?

Amendments have been introduced in conclusions to the paper.

can you provide the material allowable in terms of MPascal?

The strength of individual filaments is shown in Table 2 in MPa.

the first person (we, I...) should be avoided.

The first person (we) in the paper text was replaced with the passive mode.

Reviewer 2 Report

The manuscript presents an interesting experimental study on the formation of so-called organomorphic carbon preform based on filaments of organic precursors. The topic of the manuscript is within the scope of the journal, the results are new and interesting for the readers of the Journal of composite Science. So, my general conclusion is that the presented study deserves to be published in the journal. However, there are some key points that have to be clarified before accepting the work for publication. That is why my conclusion so far is to suggest the authors make a major revision of the text in order to address my questions and recommendations.

1. In line 121 you wrote about pressing the specimen by a quartz indenter. Please provide the dimensions of this indenter as well as its shape.

2. Figure 10 presents two images (a) and (b) but there is nothing in the text nor in the figure caption explaining what is shown in these different images. Why do you need both of these images?

3. Subsection 3.2.4 describes the properties of OCP Ipresskon that is also marked as 2D-M specimen in the text. As I understand this material is the final product of the described fabrication technology. On the other side, the term Ipresskon is firstly used on page 3 when describing a special container for thermomechanical analysis. It looks a little bit confusing. But the main question is that the authors claim that the pore diameter of the material is within the range from 4 μm to 40 μm and refer to Figure 11. But the curves in this Figure are plotted in the range from 0 to 4.5 nm. More of that, the same plots are presented in the previous paper by the author [2] and do not look as original. More of that, in Figure 12,a one can see that this material has some dense core and porous layers on the surface. Then, it is not clear which microstructure is shown in Figure 12,b. In the paragraph after Figure 12, we can also read “the density values at the surface may differ from those within the OCM volume. Therefore, the fiber strength values should also be different, as evidenced by the data shown in Table 2”. But we can not see that from Table 2, because it is not clear how the sample numbers correspond to their position in the material. More of that, I would recommend describing all characteristics presented in Table 2 (in my opinion, only the meaning of Fmax is evident).

4. Figure 13 presents tensile diagrams, but the vertical axis is labeled as Strain. It seems that it should be Force. Strain is actually a normalized deformation, so both axes in Figure 12 are deformations. Why the loading curve in Figure 12,b remains zero until deformation 0.25 %? Please explain why the specimen does not resist up to such high deformation.

5. In line 299 you refer to the tensile strength of the carbon-carbon composite given in MPa and compare it with data from Table 2 given in N without providing the cross-section area of the fibers. That is not correct.

6. Concerning the words “normalized radius” for Eq. (2) I would recommend using the term “characteristic radius” because “normalized” usually means nondimensional using the characteristic scale.

7. Please explain what is OCP in line 55 (it is written only in Keywords) and GOST 32667-2014 in line 137.

8. I would recommend thorough proofreading the text, especially in the places marked in yellow in the file attached.

Author Response

First of all, I would like to express my gratitude to the distinguished Reviewer 2 for the useful and insightfulcomments and advice that made it possible to render the manuscript better and more understandable to the reader. In the text of the new variant paper, the amendments are highlighted in green.

1. In line 121 you wrote about pressing the specimen by a quartz indenter. Please provide the dimensions of this indenter as well as its shape.

The drawing of the indenter has been included in the text of the paper (Figure 1).

2. Figure 10 presents two images (a) and (b) but there is nothing in the text nor in the figure caption explaining what is shown in these different images. Why do you need both of these images?

(Now this is Figure 11). Two images are presented, because one of them exemplifies autoadhesiveinteractions, while the other (Figure12) shows clearly the denser envelope and the less dense core the carbon fiber.

3. Subsection 3.2.4 describes the properties of OCPIpresskon that is also marked as 2D-M specimen in the text. As I understand this material is the final product of the described fabrication technology. On the other side, the term Ipresskon is firstly used on page 3 when describing a

special container for thermomechanical analysis. It looks a little bit confusing.

For better clarity, mention of Ipresskon as a preformfor the special high-density carbon-carbon container during thermomechanical analysis has been excluded from the text of the paper.

But the main question is that the authors claim that the pore diameter of the material is within the range from 4 μm to 40 μm and refer to Figure 11. But thecurves in this Figure are plotted in the range from 0 to 4.5 nm. More of that, the same plots are presented in the previous paper by the author [2] and do not look as original.

Indeed, paper [2] provides the typical data obtained using reference porometry for OCP. It is a fact that the OCP porous structure is highly reproducible, so all the curves obtained with porometry will be similar. However, it is the data obtained specifically for the preform shown in Fig.13 (a) (the new number of the figure in the revised paper) with the microstructure shown in Fig.13 (b). In Figure 14, I considered it sufficient to present only the integral pore distribution by radius within the preform. For better clarity, the name of the X axis на Figure 14 has been changed to emphasize that it shows the decimal logarithm of the characteristic radius of the pores, the size of which is expressed in nanometers(millimicrons).

More of that, in Figure 12,a one can see that thismaterial has some dense core and porous layers on the surface. Then, it is not clear which microstructure is shown in Figure 12,b. In the paragraph after Figure 12, we can also read “the density values at the surface may differ from

those within the OCM volume. Therefore, the fiber strength values should also be different, as evidenced by the data shown in Table 2”. But we can not see that from Table 2, because it is not clear how the sample numbers correspond to their position in the material.

I really appreciate the subtle remark of the distinguished Reviewer 2. Here, it should be understood that the values of the filament density must be greater near the surface of the preform than within the volume due to the absence of any diffusion obstacles for air penetration during thermal stabilization in the former case. When the density of the filaments becomes greater, given the same cross-section of the fibers and the oxidation sufficient for formation of condensed structures, its strength will also become higher. It is the notable difference in the filament strength that is emphasized in Table 2. Unfortunately, during the experiment, it was not possible technically to separate the fibers lying closer to the surface from those lying in depth.

As for Fig.12, b, it shows the uniform OCP microstructure both at the surface and in volume. The corresponding text has been added to the paper.

More of that, I would recommend describing all characteristics presented would recommend describing all characteristics presented in Table 2 (in my opinion, only the meaning of Fmax is

evident).

Definitions of all characteristics has been included in Table 2.

4. Figure 13 presents tensile diagrams, but thevertical axis is labeled as Strain. It seems that it should be Force. Strain is actually a normalized deformation, so both axes in Figure 12 are deformations. Why the loading curve in Figure 13,bremains zero until deformation 0.25 %? Please

explain why the specimen does not resist up to such high deformation.

When naming the Y axis in Figure 13 (now it is Figure 15), an error was made. Now “Strain” has been replaced with “Tensile Strength. In Figure 15, b, the deformation starts from 0.25%, rather than from zero, due to a certain initial out-of-straightness of an individual filament when pasted on to the frame. Therefore, the force will remain zero until the filament straightens (when the holders of the test machine diverge).

5. In line 299 you refer to the tensile strength of the carbon-carbon composite given in MPa and compare it with data from Table 2 given in N without providing the cross-section area of the fibers. That isnot correct.

A column showing the filament diameter values for each sample from No.1 through No.5. has been included in Table 2.

6. Concerning the words “normalized radius” for Eq. (2) I would recommend using the term “characteristic radius” because “normalized” usually means nondimensional using the characteristic scale.

“Normalized radius” for Eq. (2) has been replaced with “characteristic radius”.

7. Please explain what is OCP in line 55 (it is written only in Keywords) and GOST 32667-2014 in line 137.

OCP means “organomorphic carbon preform” obtained as a result of pyrolysis of organic (polyacrylonitrile) M, 2D-M, 2D-MS preforms in the compressed state. Line 55 refers to the carbon fibers that make up such a preform. The wording of this place in the text of the paper has been updated. In Section 2, the description of the method for determining the fiber strength has been made more complete, which made it possible to remove the mention of GOST 32667-2014.

8. I would recommend thorough proofreading the text, especially in the places marked in yellow in the file attached.

The places in the paper text indicated by the distinguished Reviewer 2 have been carefully analyzed and updated.

Round 2

Reviewer 1 Report

Dear author,

thanks for your answers. Good luck.

Reviewer 2 Report

The authors have addressed all my suggestions and questions and made corresponding changes in the current version of the manuscript.